# Positive Association between Patients’ Perception of Chronic Pain Rehabilitation as a Personally Meaningful Experience and the Flourishing Aspect of Well-Being

**DOI:** 10.3390/healthcare12161655

**Published:** 2024-08-20

**Authors:** Katrina J. Liddiard, Cary A. Brown, Annette J. Raynor

**Affiliations:** 1School of Medical and Health Sciences, Edith Cowan University, Joondalup, WA 6027, Australia; a.raynor@ecu.edu.au; 2Faculty of Rehabilitation Medicine, University of Alberta, Edmonton, AB T6G 2R3, Canada

**Keywords:** person-centred care, occupational therapy, physiotherapy, biopsychosocial, pain management

## Abstract

Chronic pain rehabilitation helps to reduce pain and restore valued life roles. Patients may have more positive outcomes when they perceive rehabilitation to be personally meaningful. This study examined associations between self-reported, personally meaningful rehabilitation and well-being. A pilot study was conducted using an online survey of people with chronic pain and experiences of rehabilitation. The PROMIS Pain Interference Short Form 8a and The Flourishing Scale were used to explore well-being. A modified self-report measure, the Meaningfulness in Rehabilitation Scale, was pilot-tested for construct validity and used in the survey. Of the 48 participants (81% female; 19% male), most attended a generalist therapy practice (62%) once per week (33%) or once per fortnight (29%). No statistically significant relationship was found between self-reported meaningfulness in rehabilitation and pain interference or other patient and therapy characteristics (duration of chronic pain category, type of therapy practice, resolution of rehabilitation category, and frequency of appointments). The nonparametric analysis identified a statistically significant moderate positive correlation between self-reported meaningfulness in rehabilitation and the flourishing aspect of well-being. This raises important questions and suggests that patients’ perception of rehabilitation as meaningful warrants further research. This pilot study provides valuable guidance to inform a larger investigation.

## 1. Introduction

Chronic pain is a condition where neurological, immune, and endocrine systems contribute to a whole-person experience [1,2]. The condition has a broad impact on the quality of life and the ability to carry out usual vocational and social functions and roles [3,4,5]. Chronic pain affects a significant part of the population, with estimates ranging from 19.2% [6] up to 51.3% [7], and this prevalence is rising [8,9]. Contemporary chronic pain management has seen a shift in priorities from a biomedical to a biopsychosocial paradigm [10,11,12], and there are now calls to extend that even further to a sociopsychobiological perspective [13] or biopsychosocial existential approach [14]. Current research suggests chronic pain management should focus more on the quality of life and well-being [15,16] rather than the singular aim of reducing pain. Rehabilitation is one strategy that helps to restore productive and meaningful engagement and improve the quality of life [17,18]. However, rehabilitation can take many different forms [13].

Chronic pain can be characterised as an ongoing state of perceived threat [19] and therefore is known to disrupt meaningful occupations and life goals [20]. Increasingly, the primary goal of rehabilitation has shifted away from merely reducing pain to a target of living well despite the pain [15,21,22]. One definition of well-being is where an individual’s biophysical, psychological, and social resources are in balance with the demands or challenges they face [23]. For many people, the demands of chronic pain far outweigh the resources they have available [24], and rehabilitation aims to redress this imbalance and increase psychological well-being [22]. People with chronic pain often either limit physical activity to maintain pain at a tolerable level or tolerate greater pain intensity to enable a level of activity that they find satisfactory [25]. Traditional measures, such as pain intensity scales, do not sufficiently capture this experience; therefore, measures of well-being better indicate the extent to which pain interferes with daily life and activities [4,15,26] and the ability to flourish in life. Flourishing is an aspect of well-being that includes perceived success with relationships; having a purposeful, meaningful, and engaged life; and contributing to others’ happiness [27].

Two rehabilitation professions that help people to manage chronic pain are occupational therapy and physiotherapy [28,29,30]. In Australia, pain management training is achieved through entry-level programs, ongoing professional development, and on-the-job experience. This means that rehabilitation therapists will adopt different approaches, depending on their training, background experiences, and their own beliefs and values [31,32]. The therapist’s approach to rehabilitation is known to influence a patient’s experience [33,34]. For example, therapy is often positively influenced by the relationship between the therapist and the patient [35]. When patients perceive their rehabilitation to be “goal oriented, meaningful and enjoyable” [36] (p. 756), better outcomes can be achieved [37]. However, in the literature on chronic pain rehabilitation, the term ‘meaningful’ is not well defined and is used in disparate ways, with a preference towards therapists’ or researchers’ standpoints rather than patients’ perspective of meaningfulness [38]. This is an important concern because healthcare professionals and patients are likely to have different views [39] on what makes rehabilitation meaningful.

There appears to be an important gap in evidence regarding the impact of a personally meaningful chronic pain rehabilitation experience on well-being. Prior to a large-scale study, an important first action is to confirm the feasibility of research design and methods [40]. Pilot studies are a foundational step to test feasibility, enable the identification of potential methodological flaws, and explore potential relationships between variables that warrant attention in the larger investigation [40,41]. This paper reports on a pilot study into the relationships between patients’ perceptions of meaningfulness in chronic pain rehabilitation and self-reported measures of well-being. The study addressed the following research questions: (1) What is the relationship between patient-reported meaningfulness in rehabilitation with patient-reported flourishing and patient-reported pain interference? (2) Is there a difference in self-reported meaningfulness in rehabilitation between groups based on age, gender, duration of pain, type of practice attended, resolution of rehabilitation, and frequency of appointments?

Additionally, the aim was to explore the potential use of a prototype scale to measure self-reported meaningfulness in rehabilitation for use in a larger study.

## 2. Materials and Methods

The aim of this study was to investigate the impact of a personally meaningful experience in rehabilitation, in particular examining associations between self-reported, personally meaningful rehabilitation and well-being. In order to achieve this aim, an operational definition of patient-defined meaningfulness in the literature on chronic pain rehabilitation was required. The definition arising from a recent concept analysis [38] was selected, and all methodological decisions were underpinned by its definition of meaningfulness in rehabilitation: “Patient-identified meaningfulness describes that which patients themselves select as being of value and relates to their personal sense of identity” [38] (p. 15).

### 2.1. Study Design

This pilot study was conducted using a quantitative, non-experimental descriptive design [42], and participants with chronic pain who had personal experience with occupational therapy, physiotherapy, or both, were recruited. The study was conducted in accordance with the Declaration of Helsinki of 1964, and approval was granted from the relevant institutional Human Research Ethics Committee [approval no. 21008; 20 November 2019] prior to recruitment. Informed consent was obtained from all participants, and responses were anonymous.

### 2.2. Participants

From February to October 2020, Australian adults with chronic pain were recruited through social media; posters displayed on university campuses; and email via the primary researcher’s therapy colleagues and coworkers in therapy practices and specialist pain management practices. Participants were included if they reported personal experience of chronic pain; had experience of physiotherapy or occupational therapy within the past 0–52 weeks; and were over 18 years of age. Exclusion criteria included those with significant cognitive impairment or non-English speaking and those actively attending rehabilitation at the time.

### 2.3. Procedure

The Qualtrics XM Platform was used to construct and distribute the survey and collect data for analysis. Participant information was provided at the beginning of the survey, and consent was assumed if participants continued into the survey. The anonymous survey opened with questions regarding demographic information such as age and gender, as the evidence suggests that the experience of chronic pain may vary with these [43,44]. Based on the authors’ hypothesis that meaningfulness in rehabilitation may differ according to other patient and therapy characteristics, such as duration of pain [45] or type of therapy practice [46], these were also included. Participants then responded to three self-report measures to capture pain interference [47], flourishing [27], and meaningfulness in rehabilitation, using the scales described below.

### 2.4. Scales

#### 2.4.1. Pain Interference

The Patient-Reported Outcomes Measurement Information System (PROMIS) Pain Interference Short Form 8a [47] was used to measure self-reported pain interference. This is a psychometrically tested (α = 0.99) measure of the impact of pain on engagement in emotional, social, physical, and recreational activities [47]. The scale rates eight statements on the extent to which pain interferes with day-to-day activities; social participation; and activities usually considered fun, on a 5-point, Likert-type scale from 1 (“not at all”) to 5 (“very much”). The eight responses are then summed to produce a single pain interference score between 8 and 40, with a high score indicating greater pain interference.

#### 2.4.2. Flourishing

The Flourishing Scale [27] is a standardised self-report measure of eight items relating to aspects of flourishing such as optimism, self-esteem, purpose, and relationships [27,48]. The scale has good psychometric properties with high internal consistency (α = 0.87) [27] and reflects aspects of both hedonic well-being (which relates to a sense of pleasure, positive affect, and life satisfaction) and eudaimonic well-being (which relates to functioning well and a sense of meaning in life) [49]. Examples of statements include ‘I lead a purposeful and meaningful life’; ‘I actively contribute to the happiness and well-being of others’; and ‘I am optimistic about my future’ [27]. Participants rate eight statements on a 7-point, Likert-type scale from 1 (“strongly disagree”) to 7 (“strongly agree”), summed to produce a single score between 8 and 56. A high score indicates greater psychological strengths and resources.

#### 2.4.3. Meaningfulness in Rehabilitation

After an extensive search with guidance from a medical librarian, no existing measure of patient-defined meaningfulness in rehabilitation was found. It was determined that a new measure would be required. Throughout this process, all decisions were informed by the definition of meaningfulness in rehabilitation previously cited from the concept analysis [38]. A measure that closely aligned with the concept of meaningfulness in rehabilitation was identified; modified, with permission from the lead author [50,51]; and piloted to confirm construct validity [52]. The Meaningfulness in Songwriting Scale (MSS) was developed by Baker, Silverman, and MacDonald [50,51] as a measure of affective, cognitive, and relational dimensions of meaningfulness, in the process and product of therapeutic songwriting. The MSS has been psychometrically tested in several settings including an acute psychiatric population, demonstrating strong internal consistency (α = 0.98) and acceptable test–retest reliability (ICC2,1 = 0.93) [51].

The language of the MSS was adapted to create the Meaningfulness in Rehabilitation Scale, and the 21 items were pilot-tested to confirm construct validity, with a convenience sample of seven participants (four females, three males; aged 18 to 65), who were not undergoing chronic pain rehabilitation. The piloting process determined that 13 of the total 21 modified statements more accurately reflected the experience of rehabilitation rather than the outcome of rehabilitation. Correspondence with the primary author of the MSS [51] confirmed that these 13 items were designed to measure meaningfulness in the process of therapeutic songwriting rather than the meaning derived from the product or songwriting output. Based on this information, the 13-item scale was deemed a better fit with the concept of meaningfulness in rehabilitation. The final version of the Meaningfulness in Rehabilitation Scale (MRS13) rated 13 statements on a 5-point, Likert-type scale from 1 (“strongly disagree”) to 5 (“strongly agree”). Scores were added to provide a single score between 13 and 65. A high score indicated greater perceived meaningfulness in rehabilitation.

### 2.5. Statistical Analysis

All analyses were performed with the statistical program SPSS version 29.0 (SPSS Inc., Chicago, IL, USA), with *p* values < 0.05 considered statistically significant. Demographic data (gender; age group) of the sample and the details of the individual’s pain and treatment (duration of chronic pain, type of practice attended, how rehabilitation was resolved, and frequency of appointments) are presented using descriptive statistics (frequencies and percentages).

The Meaningfulness in Rehabilitation Scale (MRS13), the Flourishing Scale (FS8), and the PROMIS Pain Interference Short Form 8a (PI8a) all use Likert-type responses. It is acknowledged that some debate exists in healthcare about whether to treat data derived from Likert-type scales as continuous or ordinal [53]. In this instance, scores from all three scales were treated as continuous variables [54], based on their use in the originating studies [27,47,51]. Spearman’s rank correlation [55,56] was used to test the correlations between self-reported meaningfulness in rehabilitation (MRS13) and the dependent variables of self-reported pain interference (PI8a) and self-reported flourishing (FS8). The strength of these correlations was reported [55]. The differences between groups of MRS13 scores were calculated using Kruskal–Wallis or Mann–Whitney U tests for patient and treatment characteristics (duration of chronic pain category, type of therapy practice, resolution of rehabilitation category, and frequency of appointments) [56].

## 3. Results

Surveys were completed by 68 people. Incomplete outcome measures for 20 participants meant that, in total, 48 surveys were analysed [Table 1]. The spread of ages was fairly broad (Mdn = 46; SD = 14.17; range 21 to 74 years; 18–29 years = 8; 30–39 years = 8; 40–49 years = 10; 50–59 years = 13; 60–69 years = 8; 70+ years = 1), and the majority of participants had a chronic pain duration of more than five years (56.3%). Most attended a generalist therapy practice (62.5%) or did not know the nature of the practice (22.9%), with only a small number selecting specialist pain practice (14.6%) [Table 1].

Shapiro–Wilk tests for normality showed that the Meaningfulness in Rehabilitation Scale (MRS13) scores were normally distributed (W(48) = 0.979, p = 0.532). By contrast, the Flourishing Scale (FS8) scores (W(48) = 0.936, *p* ≤ 0.012) and the PROMIS Pain Interference Short Form 8a (PI8a) scores (W(48) = 0.876, *p* ≤ 0.001) were not normally distributed; therefore, nonparametric tests were used for all statistical analyses.

A Spearman’s Rho test showed a statistically significant, moderate positive correlation between MRS13 and FS8 scores [Table 2], indicating that participants who scored higher for personally meaningful rehabilitation also reported greater flourishing. Spearman’s Rho tests also showed a weak negative correlation between MRS13 and PI8a scores and a weak positive correlation between MRS13 and age; however, neither finding was statistically significant [Table 2].

Based on Mann–Whitney U or Kruskal–Wallis tests, there were no statistically significant differences between groups in MRS13 scores relative to all other patient and treatment characteristics (gender, duration of chronic pain, type of therapy practice, resolution of rehabilitation category, and frequency of appointments) [Table 2].

## 4. Discussion

This pilot study provides valuable insight and foundational proof to justify further research into what makes rehabilitation meaningful from the patient’s perspective. Future work, including the research design and methods required for a larger study, will be informed by these results. With regard to the first research question, there was a statistically significant, moderate positive correlation between self-reported meaningfulness in rehabilitation and the flourishing aspect of well-being. This provides preliminary evidence that rehabilitation perceived by the patient to be personally meaningful may be associated with an important outcome related to living well with chronic pain. This pilot study found no evidence of an association between meaningfulness in rehabilitation and pain interference; however, further study into this finding is also warranted. A qualitative study [57] (Liddiard et al., 2022), which was completed concurrently, indicated that meaningfulness has idiosyncratic aspects depending on the individual. Common themes identified in that study suggest that clients view meaningful rehabilitation as having “a *genuine connection* with a *credible therapist*, who can act as a *guiding partner* to address what the client *self-defines* as *personally valued*, and *relevant to their self-identity*” [57] (p. 696). Future research will consider findings from both studies.

The results of this study do not imply causality in the correlation between meaningfulness in rehabilitation and the flourishing aspect of well-being; however, they raise salient questions about the nature of personally meaningful rehabilitation and how it may relate to the flourishing aspect of well-being. The operational definition that was used to develop the MRS13 scale was based on the definition of meaningful rehabilitation as self-determined by the patient in a personally valued direction and relevant to their sense of self-identity [38]. The flourishing aspect of well-being measured by the FS8 is characterised by Diener et al. [27] as a perception of a purposeful, meaningful, and engaged life; contributing to the happiness of others; optimism and self-respect; connection through social relationships; and competence in valued activities [27]. It is plausible that a person who is engaged in this way with their life could be inclined to either seek or perceive a more personally meaningful experience within rehabilitation. Equally, a patient who is encouraged, in a personally meaningful rehabilitation experience, to self-determine their own personally valued goals and targets that contribute to their sense of self-identity might naturally perceive themselves to be flourishing in life. This pilot study provides support to investigate this relationship and explore the possible directionality in more detail after the MRS13 tool has been further developed and psychometrically tested.

The experience of pain is multi-faceted [58,59], and this pilot study reinforces the need to investigate patients’ experiences of meaningfulness in rehabilitation in greater depth. Recent research has shown that when therapy goals are underpinned by a person’s global meaning or sense of meaning in life, the person is also likely to derive greater situational meaning or the sense that their therapy goals are meaningful [60,61]. Rehabilitation that the patient is able to self-direct towards personally valued goals that are relevant to their sense of self-identity would theoretically see them engage with their valued and meaningful occupations outside of therapy. Engaging with valued occupations is understood to contribute to feelings of contentment, identity, competence, autonomy, and belonging [62], which are also closely aligned with the flourishing aspects of well-being such as optimism, self-esteem, purpose, and relationships [27,48]. If therapy is centred around activities and goals that are coherent with the patient’s sense of self-identity, this may help to address one of the core challenges of learning to live with a chronic condition, which is the attempt to find (or re-find) meaning in life [63]. Some people are severely challenged by chronic pain, whereas others experience well-being [22], and it is not yet clear whether a personally meaningful rehabilitation experience is a contributing factor to this difference.

Future investigations will need to consider potential moderating influences and examine whether the correlation between personally meaningful rehabilitation and flourishing is a direct relationship or is moderated by well-established constructs such as person-centeredness [64,65] or the therapeutic relationship [66,67]. Person-centredness has been shown to result in positive outcomes [15]. Additionally, a stronger therapeutic relationship [68] may lead to greater trust and connection [69] and positive well-being benefits [70]. Despite a strong movement to shift the power dynamic in healthcare over the past few decades, paternalistic attitudes and behaviours still remain [71], and some patients may feel they are not entitled to direct their own healthcare [31]. Personally meaningful rehabilitation sits in contrast to this. It represents an experience of rehabilitation where the person is able to self-determine what they personally value in therapy and consider relevant to their sense of self-identity. This experience may signal to the patient that the therapist not only values their perspective but sees it as central to the process of rehabilitation. It remains to be seen whether the person-centredness of this approach, or the therapeutic relationship that likely develops out of it, are moderating factors in the positive correlation between personally meaningful rehabilitation and flourishing.

The relationship between meaningfulness in rehabilitation and pain interference is potentially more nuanced, and the scale of this pilot study possibly contributed to the findings. Despite this finding, the authors believe pain interference is a relevant part of the picture. Some people are known to live with a sense of well-being despite pain interfering with their daily lives [22]. For example, a person can find life satisfying, and their job rewarding, despite the pain caused by computer-based tasks. Whilst pushing through the pain then interferes with their daily activities, they continue to experience a sense of living well [25]. The experience gained from this pilot study will help to refine the research design, elucidate the relationship between pain interference and meaningfulness in rehabilitation, and identify other important measures of living well with chronic pain. A larger study will provide an opportunity to examine pain interference and associated constructs in greater depth.

The authors hypothesised that factors such as attendance at a specialist practice and the frequency of appointments could contribute to an experience of therapy as meaningful. In regard to the second research question, no statistically significant difference between groups was identified with any of the patients’ pain or therapy characteristics and the meaningfulness in rehabilitation scores [Table 2]. A reasonable hypothesis was that the experience of rehabilitation as personally meaningful may be reflected in the circumstances of how therapy was resolved [72]. The scale of this pilot study may have contributed to the lack of statistically significant findings; however, the way therapy was ceased, and by whom, may be more closely aligned to other constructs such as patient satisfaction. This pilot study highlights an important need to refine the parameters for subsequent research. One interesting finding that emerged was that the majority of participants attended a generalist practice (62.5%) or did not know what type of practice they attended (22.9%). With the limited availability of specialist practices and the rising prevalence of chronic pain, it is likely that many people with chronic pain will continue to see generalist therapists for rehabilitation. If no difference with specialist therapist intervention exists, then it would make sense to focus education efforts on upskilling and equipping therapists more broadly to enable personally meaningful rehabilitation for the wider population of people with chronic pain. There is a responsibility then to determine whether generalist therapists can become adept in delivering a personally meaningful form of rehabilitation for people with chronic pain. This reinforces the need for allied health professionals to be trained in the most effective chronic pain management approaches [73], and how best to address this need will be an important part of future research into personally meaningful chronic pain rehabilitation.

The secondary aim was achieved, i.e., to explore the potential use of a prototype scale to measure self-reported meaningfulness in rehabilitation. The pilot study findings support the case for further research, in particular tool development and testing. The MRS13 requires considerable research in order to refine it and test psychometric proprieties; however, given the positive relationship between flourishing and meaningfulness, more research is warranted. In most parts of the public and private health system, health professionals are under significant time pressure often driven by cost constraints [74]. If therapists and researchers better understand the impact of a personally meaningful experience of chronic pain rehabilitation, they may be more inclined to prioritise this approach despite this pressure. A psychometrically tested tool to evaluate this aspect of chronic pain rehabilitation will be valuable in this endeavour.

As with all research, there were limitations. Recruitment was carried out during the COVID-19 pandemic. As people were encouraged to work from home in parts of Australia, a lack of desire to be online during discretionary time may have slowed and limited the size of the sample. Despite this, sufficient data were collected to provide valuable insight into the design and implementation of intended future research. Participants were not tested to enforce the cognitive capability exclusion criterion, and this should be considered as a potential limitation.

## 5. Conclusions

This pilot study explored the gap in evidence about the self-reported experience of meaningfulness in rehabilitation and important well-being outcomes. It demonstrated feasibility for a larger study and provided valuable insight into refining the design of future research. Further development of a specific psychometrically tested measure of meaningfulness in rehabilitation is warranted. The finding of a positive correlation between meaningfulness in rehabilitation and the flourishing aspect of well-being is encouraging and supports further investigation into what makes the experience of rehabilitation personally meaningful for people with chronic pain.

## Figures and Tables

**Table 1 healthcare-12-01655-t001:** Demographic data.

Demographics (*n* = 48)	Frequency (%)
Gender: *	
female	39 (81%)
male	9 (19%)
Duration of chronic pain:	
6–12 months	2 (4.2%)
1–5 years	19 (39.5%)
5–10 years	6 (12.5%)
>10 years	21 (43.8%)
Type of practice attended:	
specialist	7 (14.6%)
generalist	30 (62.5%)
unknown	11 (22.9%)
How rehabilitation was resolved #:	
therapist considered resolved	4 (8.3%)
therapist unable to resolve	4 (8.3%)
I considered resolved	2 (4.2%)
I felt not improved	14 (29.2%)
other	24 (50%)
Frequency of appointments:	
more than once/week	8 (16.7%)
once/week	16 (33.4%)
once/fortnight	14 (29.1%)
once/month	3 (6.3%)
other	7 (14.6%)

* The gender option of non-binary was offered but not selected. # Other reasons given for treatment cessation included therapy interrupted by COVID-19; lack of further funding; continuing treatment in different forms, e.g., home exercises or remedial massage; mutual decision between therapist and patient; and condition worsened and did not wish to continue.

**Table 2 healthcare-12-01655-t002:** Statistical findings.

Independent Variable	Dependent Variable	Results
Gender ^1^	MRS13	Mann–Whitney U test showed no significant difference between the MRS13 scores of females and males (U = 200.500 OR z = 0.661, *p* = 0.515)
Duration of chronic pain ^2^	MRS13	Kruskal–Wallis test showed no significant difference in MRS13 scores between chronic pain duration categories #, χ2 (3, *n* = 48) = 0.480, *p* = 0.923
Type of therapy practice ^3^	MRS13	Kruskal–Wallis test showed no significant difference in MRS13 scores between practice categories ^3^, χ2 (2, N = 48) = 2.977, *p* = 0.226
Frequency of appointments ^4^	MRS13	Kruskal–Wallis test showed no significant difference in MRS13 scores between frequency of appointment categories ^4^, χ2 (4, N = 48) = 1.108, *p* = 0.893
Resolution of rehabilitation ^5^	MRS13	Kruskal–Wallis test showed no significant difference in MRS13 scores between resolution of rehabilitation categories ^5^, χ2 (4, N = 48) = 6.948, *p* = 0.139
Age	MRS13	Spearman’s Rho test showed a weak positive correlation between age and MRS13 score that was not statistically significant, rs(46) = 0.062, *p* = 0.675
MRS13 ^6^	PROMIS Pain Interference 8a ^7^	Spearman’s Rho test showed a weak negative correlation between PI8a score and MRS13 score that was not statistically significant, rs(46) = −0.10, *p* = 0.489
MRS13	Flourishing Scale ^8^	Spearman’s Rho test showed a moderate positive correlation between FS8 score and MRS13 score that was statistically significant, rs(46) = 0.405, *p* = 0.004

^1^ Gender = male/female (non-binary offered but not selected). ^2^ Duration of chronic pain = (categories < 3 months and 3 to 6 months were not selected; 6 to 12 months; 1 to 5 years; 5 to 10 years; >10 years). ^3^ Type of therapy practice = (specialist; generalist; unknown). ^4^ Frequency of appointments = (>1x/week; 1x/week; 1x/fortnight; 1x/month; other). ^5^ Resolution of rehabilitation = (‘therapist considered resolved’; ‘therapist unable to resolve’; ‘I considered resolved’; ‘I felt not improved’; other). ^6^ MRS13 score = total score between 13 and 65. ^7^ Pain Interference Short Form 8a score = total score between 8 and 40. ^8^ Flourishing Scale score = total score between 8 and 56.

## Data Availability

The raw data supporting the conclusions of this article will be made available by the authors upon request.

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
