# Peer review of "Positive Association between Patients’ Perception of Chronic Pain Rehabilitation as a Personally Meaningful Experience and the Flourishing Aspect of Well-Being"

_healthcare, 2024, doi:10.3390/healthcare12161655_

Round 1

Reviewer 1 Report

Comments and Suggestions for Authors

I am grateful for the invitation to review this manuscript. As the authors rightly conclude, this study raises important questions and highlights the relevance of patients' perception of rehabilitation as meaningful. However, these results may be impractical if the rehabilitation process and its approach are not well described. For example, was it a biopsychosocial approach to pain management? Were the physiotherapy or occupational therapy interventions passive (manual therapy, electrotherapy, kinesiotaping, orthoses) or active therapies that encourage self-management of pain (pain education and behavioural changes, exercise)? In how many cases was the intervention multimodal? In how many cases was the intervention transdisciplinary? Further exploration of these issues could significantly increase the impact of this study. If you decide to include these topics in your manuscript, you should also mention them in the introduction. Here is a bibliography that you may find useful.

Lin I, Wiles L, Waller R, et al. What does best practice care for musculoskeletal pain look like? Eleven consistent recommendations from high-quality clinical practice guidelines: systematic review. Br J Sports Med. 2020;54(2):79-86.

Lin I, Wiles L, Waller R, et al. Patient-centred care: the cornerstone for high-value musculoskeletal pain management. Br J Sports Med. 2020;54(21):1240-1242.

Lewis JS, Stokes EK, Gojanovic B, et al. Reframing how we care for people with persistent non-traumatic musculoskeletal pain. Suggestions for the rehabilitation community. Physiotherapy. 2021;112:143-149.

The, here are some specific details about this manuscript:

-Introduction:

I suggest rewriting the last paragraph on the research question to make it more fluid. For example: "The study addressed the following two research questions: (1) What is the relationship of patient-reported meaningfulness in rehabilitation with patient-reported flourishing and patient-reported pain interference?; (2) Is there a difference in self-reported meaningfulness in rehabilitation between groups based on age; gender; duration of pain; type of practice attended; resolution of rehabilitation and frequency of appointments?"

- Methods

Please provide more details on the inclusion criteria. What was the definition of chronic pain (symptoms persisting for more than three months?); Does it refer to chronic musculoskeletal pain? or neuropathic pain? or oncological pain? or postsurgical pain?; Was any particular location specified (spine, knee, shoulder)? Also, if the authors have this information, it would be good to add it to Table 1. In addition, as mentioned above, add a section to describe in depth the characteristics of the rehabilitation.

- Results

Add age data to Table 1. 

-Discussion
The discussion of the manuscript should also be enriched by following the changes suggested above.

Author Response

Thank you for the helpful and constructive feedback. Please see the attachment.

Reviewer 2 Report

Comments and Suggestions for Authors

This is a review of the paper entitled, “Positive association between patients perception of chronic pain rehabilitation as a personally meaningful experience, and the flourishing aspect of well-being” [Healthcare-3037672]. The paper is a descriptive study Participants in rehabilitative therapies (physiotherapy and occupational therapy) while undergoing treatment for their chronic painful conditions. This is an important topic area, as for far too long patient voices and perspectives have not been considered in selection of treatment outcomes. The paper may be improved by consideration of the following feedback.

Introduction: While the introduction seems appropriate with relevant literature cited, The paper could be improved by including other research literature that suggests the importance of meaningfulness of clinical outcomes in pain trials.

Page 2, lines 85-87. Comparing differences among this many groups may require an estimation of power to detect true differences, especially paying attention to multiple comparisons and the importance of correcting for those multiple comparisons. Please provide an estimation of power to detect true differences and rationale for not correcting for multiple comparisons.

Page 2, lines 88-89. The authors indicate exploration of the feasibility of using the prototypic scale. How was feasibility measured?

Methods

Page 3, line 147: it would be helpful for the reader if an alpha value were provided for this particular scale. Please share that information.

Page 4, lines 160 through 173:

Please help the reader understand the qualifications of the convenience sample of seven participants. Were they patients? Or did they have content expertise in the rehabilitation process? More information is needed here.

It may be helpful to provide some examples of the scale items. Were there other psychometric qualities that were used in determining which items held together? For example was a factor analysis performed? If so, please provide results.

It seems appropriate that the scale would be pilot tested among individuals undergoing pain rehabilitation treatment. Psychometric qualities of the scale are vitally important to be reported in this initial stages of research with it. This is particularly true since it was heavily adapted from an entirely different field.

Results

Page 5, table one: it appears that the formatting may be off a little bit and the lines not always lining up with the results, please check.

Page 5, Table 1: how was the item “how rehabilitation was resolved closed quote asked, by whom? Please provide more details

How do the authors interpret the fact that most people did not find rehabilitation helpful or finished without resolution? Could this influence interpretations of the results?

Page 6, Table 2: this table would benefit from reformatting as it is rather confusing in its current form. It may make sense to list only the independent variable, labeling it variable as the first column then list the test statistic denoting statistically significant comparisons with asterisks in subsequent column. It may read better if you remove text such as “Mann-Whitney U test showed no significant...”

Page 6, line 233: given the exploratory nature of this study, it may make sense to create an entirely separate table with the statistics (mean, range and standard deviation) of each of the scales, especially focusing on the MRS13. The statistics related to the MRS13 seem vitally important to establish the reliability and some of the validity of this newly formed scale.

Discussion

this section of the manuscript needs significant reductions and reformatting. It may help to scale back the conclusions that are drawn, stating realistic conclusions based on the evidence gathered and displayed in this paper. For example, “This pilot study provides valuable insight, and foundational proof of concept...” The authors may want to qualify this or eliminate the Statement as it is unclear what proof was shared to support this statement.

Page 7, lines 256-280 -This literature review and overview would best be placed in the introduction as it helps make the case for examining personal meaning in rehabilitation.

Overall, this paper would benefit from more psychometrically sound methodology to determine the qualities of the meanings in relationships scale itself prior to piloting it to compare groups of individuals or even correlations between other constructs. More detail is warranted in describing how the items of the new scale were derived. Additional statistical analysis to determine empirically how th the scales items relate to one another would also be necessary for readers to draw conclusions about what the scale is measuring. Changes to the discussion section may also help the manuscript be more susinct and clear.

Author Response

(The authors gave the same response as above.)

Reviewer 3 Report

Comments and Suggestions for Authors

Dear authors:

I reviewed your paper entitled “Positive association between patients’ perception of chronic pain rehabilitation as a personally-meaningful experience, and the flourishing aspect of well-being”.

The study aims to explore the feasibility of using a prototype scale to measure self-reported meaningfulness in rehabilitation for use in a larger study to accomplish it, it was developed a pilot study using an online survey of Australian people with chronic pain who had personal experience of occupational therapy or physiotherapy. The PROMIS Pain Interference Short Form 8A; and The Flourishing Scale were used to explore well-being. A modified self-report measure, the Meaningfulness in Rehabilitation Scale. Participated 48 people with chronic pain. The authors concluded that this pilot study provides valuable guidance to inform a larger investigation in the area.

I would like to congrats the authors, it is very interesting study. The chronic pain is a huge problem around the world and should be applied all possible efforts to achieve an efficient control, to provide the best QoL to those who suffer with chronic pain, and them caregivers. I believe the readers of the journal will read it with the same interest I did. It was easy to read the paper.

I have some suggests to reflection and improve your paper:

·        In abstract: the aim should be the same described in page 2 lines 88-89.

·        In page 3: you refer as exclusion criteria: people “with significant cognitive impairment”, did you use some scale to measure it? How you did?

·        In page 3 “Statistical analysis” could be interesting to add the reason why you chose non-parametric statistics, instead of page: 5 lines 200-204.

It was delighting read your paper. I hope read future papers from you.

I have nothing to add, and I wish you good luck towards publishing it!

Best regards.

Author Response

(The authors gave the same response as above.)

Round 2

Reviewer 1 Report

Comments and Suggestions for Authors

I thank the authors for carefully addressing each of my comments.

I now have only a few minor details that need to be revised:

-Line: 194-195: Add "participants" after each interval, e.g., 18-29 years = 8 participants. Since it is not clear

-Line 250: please remove "(p.696)"